# Approximate SU(4) spin models on triangular and honeycomb lattices in twisted AB-Stacked WSe$_2$ homobilayer

Shuchen Zhang,[1, 2, *] Boran Zhou,[1, †] and Ya-Hui Zhang[1, ‡]

[1]*William H. Miller III Department of Physics and Astronomy,*
*Johns Hopkins University, Baltimore, Maryland, 21218, USA*
[2]*Department of Physics, Massachusetts Institute of Technology, Cambridge, Massachusetts, 02139, USA*

(Dated: February 16, 2023)

In this paper, we derive lattice models for the narrow moiré bands of the AB-stacked twisted WSe$_2$ homobilayer through continuum model and Wannier orbital construction. Previous work has shown that an approximate SU(4) Hubbard model may be realized by combining spin and layer because inter-layer tunneling is suppressed due to spin $S_z$ conservation. However, Rashba spin-orbit coupling (SOC) was ignored in the previous analysis. Here, we show that a Rashba SOC of reasonable magnitude can induce a finite but very small inter-layer hopping in the final lattice Hubbard model. At total filling $n = 1$, we derive a spin-layer model on a triangular lattice in the large-U limit where the inter-layer tunneling contributes as a sublattice-dependent transverse Ising field for the layer pseudospin. We then show that the $n = 2$ Mott insulator is also captured by an approximate SU(4) spin model, but now on honeycomb lattice. We comment on the possibility of a Dirac spin liquid (DSL) and competing phases due to SU(4) anisotropy terms.

## I. INTRODUCTION

Recent experimental progresses in moiré superlattices[1–22] make it possible to study spin physics in a Hubbard model on a moiré superlattice[23–31]. With these highly tunable systems, it is natural to ask whether we can realize unconventional spin phases which are not magnetically ordered. One route to disordering magnetic order is through adding frustration[32–40] or charge fluctuation[41, 42] to a spin 1/2 model. Another route is to consider an SU(N) spin model with $N > 2$, where the magnetic order is generally suppressed and a spin liquid phase may be stabilized even in the simplest Heisenberg model[43, 44]. However, it is challenging to realize an SU(N) spin model in real systems. Graphene moiré system may simulate an SU(4) spin model[45]. We also note a recent proposal of SU(8) model in twisted bilayer graphene[46]. One issue for the graphene moiré systems is that there is generically a large valley contrasting flux[47]. In this paper, we focus on the moiré superlattices formed by transition metal dichalcogenides(TMDs), where the valley contrasting flux is usually negligible[23], except in twisted AA stacked TMD homo-bilayer with a large displacement field[26].

With TMD layers, one can realize a moiré bilayer to simulate an SU(4) Hubbard model by combining the spin and layer degree of freedom[48]. If the inter-layer tunneling is suppressed and inter-layer distance is much smaller than the moiré lattice constant, there is an approximate SU(4) symmetry. There are two different ways to suppress the inter-layer tunneling to realize a moiré bilayer. The first one is to insert an insulating barrier such as a hexagon boron nitride(hBN) layer in the middle. However, a thick hBN will lead to a smaller inter-layer repulsion compared to the intra-layer repulsion and reduces the U(4) symmetry to $U(2) \times U(2)$[49]. Although interesting phases can still arise in this less symmetric case[49], in this paper we are interested in improving the SU(4) symmetry. The second

realization of a moiré bilayer is the twisted AB-stacked transition metal dichalcogenide (TMD) homo-bilayer[48], which was recently realized in an experiment[50]. AB stacking is generated from the standard AA stacking through rotating one TMD layer by 180° relative to the other layer. Therefore, the two TMD layers at the same valley now have opposite spin splittings. The highest valence bands have opposite spin $S_z$ in the two layers at the same valley (see Fig.1(a)). As a result, the inter-layer tunneling is forbidden due to the spin $S_z$ conservation[51].

However, in the analysis above, we ignore the Rashba spin-orbit coupling (SOC). With a Rashba SOC, spin $S_z$ is not a good quantum number anymore and one may imagine a finite inter-layer tunneling, similar to what is happening in the AB-stacked TMD hetero-bilayer[52, 53]. In this paper we analyze the effect of the Rashba SOC in the AB stacked TMD homo-bilayer. It is known that Rashba SOC usually arises in hetero-structures with a finite vertical electric field[54]. This is indeed expected for TMD hetero-bilayer. However, for TMD homo-bilayer, there is no vertical electric field if the displacement field $D$ is zero. Hence, the Rashba SOC may be suppressed. When $D = 0$, there is indeed a symmetry $C_{2x}$ which exchanges the two layers and forbids the layer symmetric Rashba SOC. The only allowed Rashba SOC term must be opposite in the two layers. It is unclear to us how this layer-opposite Rashba SOC can arise microscopically. We conjecture its value is negligible. But to obtain an upper bound of the inter-layer tunneling, we simply add layer-opposite Rashba SOC of a large magnitude (comparable to the value in a hetero-bilayer) to the continuum model and then extract an on-site inter-layer hopping term in low-energy lattice Hubbard model through Wannier orbital construction. The inter-layer hopping turns out to be quite small (at order of $0.001t - 0.01t$) even with large layer-opposite Rashba SOC.

At total filling $n = 1$, the Mott insulator in the large-U limit is described by an approximate SU(4) model. The small inter-layer hopping results in a sublattice dependent transverse Ising field in the layer pseudospin space in a convenient gauge. At zero magnetic field, previous work showed that the ground state of the SU(4) Heisenberg mdoel on a triangular lattice has a plaquette-ordered ground state[51].

* These authors contributed equally.; szhan106@jhu.edu
† These authors contributed equally.; bzhou12@jhu.edu
‡ yzhan566@jhu.edu

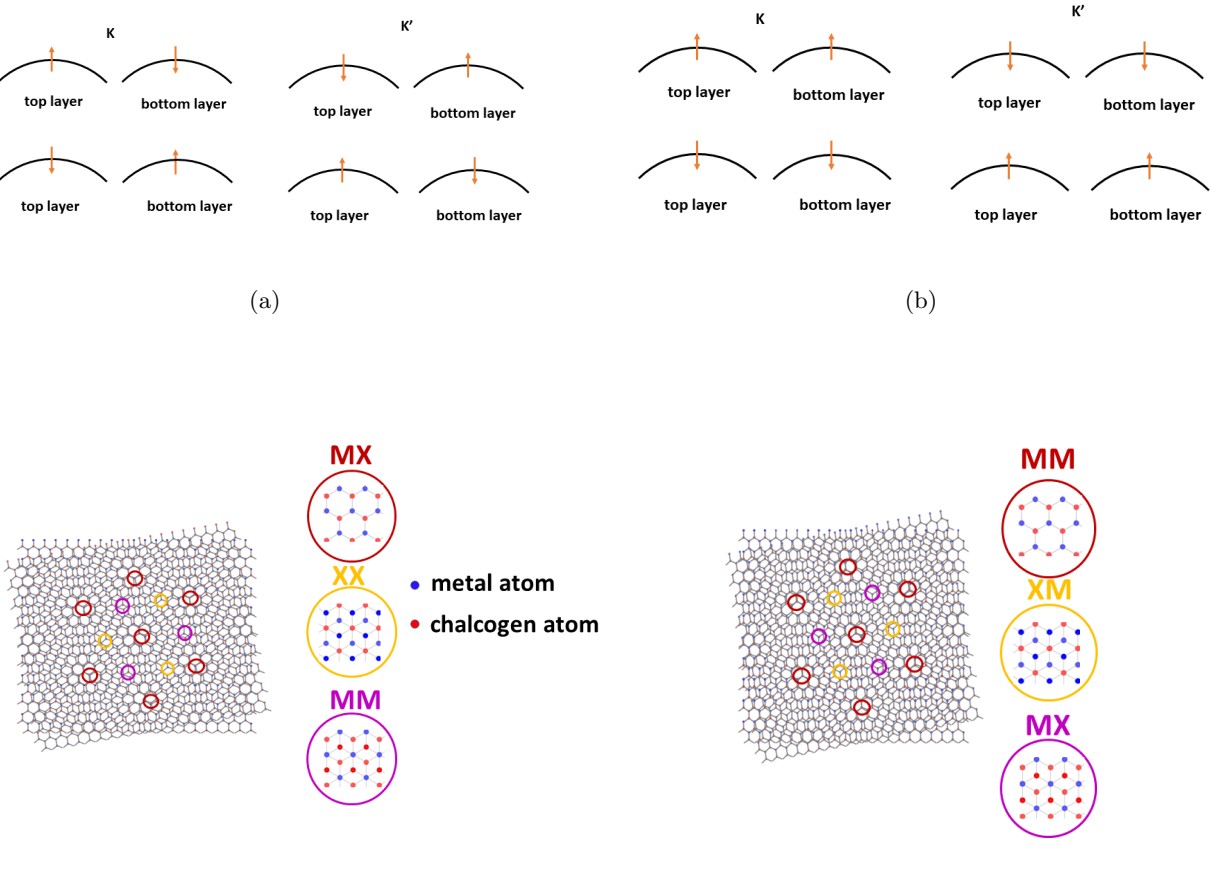

FIG. 1: (a) Illustration of spin-valley locking of the low-energy valence bands in AB stacking. (b) Illustration of spin-valley locking of the low-energy valence bands in AA stacking. (c) Schematics of the moiré pattern formed by AB-stacked twisted bilayer TMD M$X_2$. High symmetry regions are highlighted with circles. 'MX' means the metal atoms in the top layer are aligned with the chalcogen atoms in the bottom layer, and other areas are defined similarly. There is a $C_{2x}$ symmetry which rotates the system by 180° around the x axis. (d) Moiré pattern of AA-stacked twisted bilayer $MX_2$. There is a $C_{2y}$ symmetry now.

We expect that the plaquette-ordered phase is stable with a small transverse Ising field. We can polarize the real spin with a large magnetic field in the z-direction and get a spin-1/2 XXZ model for the layer pseudospin[55]. We analyze the phase diagram with an additional displacement field and a small sublattice dependent transverse Ising field originating from the inter-layer tunneling. We find a 1/3 plateau in layer polarization when varying the displacement field $D$, which is stable against the small transverse Ising field.

We then turn to the total filling $n = 2$. Naively, one may expect a Mott insulator on a triangular lattice where each site is occupied by two particles, resulting in a $SO(6)$ representation of the approximate $SU(4)$ symmetry[45]. However, a careful analysis shows that the two particles in a unit cell prefer to stay in A, B sublattice of a honeycomb lattice to reduce the on-site Hubbard U[25]. An appropriate lattice model for $n > 1$ turns out to be an approximate U(4) Hubbard model on a honeycomb lattice, which we derive through Wannier orbital construction of the lowest two moiré bands (each band consists of four flavors coming from valleys and layers). Then the Mott insulator at $n = 2$ is described by an approximate SU(4) spin model on a honeycomb lattice, which may host a Dirac spin liquid as indicated by previous

calculations[56–58] and theory[59]. We discuss the effect of anisotropy terms and possible competing phases.

## II. CONTINUUM MODEL WITH RASHBA SOC

### A. Continuum model and symmetry

We consider a twisted TMD homo-bilayer with a small twist angle $\theta$. Due to the lack of inversion symmetry within each TMD layer, there are two inequivalent stacking patterns at zero twist angle. In the AA stacking, each atom of one TMD layer aligns with the corresponding one in the other layer. In the AB stacking, one rotates one of the TMD layers by 180° relative to the other. In each TMD layer, it is known that the valence bands from the two spins $S_z = \uparrow, \downarrow$ have opposite splittings in the two valleys $K, K'$ of the hexagon Brillouin zone(BZ). In the AB stacking, the definition of the valley in one layer is flipped compared to that in the AA stacking due to the 180° rotation. Hence in the AB stacking, the two layers have opposite spin $S_z$ at the same valley (see Fig.1(a)(b)) for the same band. As a consequence, the inter-layer tunneling in the AB stacking is forbidden due to

the spin $S_z$ conservation. This property still holds at a small twist angle $\theta$ relative to the AB stacking. Another difference between the twisted AB stacking and AA stacking TMD homo-bilayer is in their symmetries. AA stacking has a $C_{2y}$ symmetry at zero displacement field which rotates the system by 180° around the y-axis and exchanges the two layers, while AB stacking's corresponding symmetry is $C_{2x}$ which rotates the system by 180° around the x-axis and exchanges the two layers(see Fig.1(c)(d)). As we will see, the different symmetries in the two stacking patterns lead to different Hamiltonians and properties.

Due to the twist, the valence bands from the valley $K$ and $K'$ are folded into a mini moiré Brillouin zone(MBZ) as shown in Fig.2. We construct a continuum model including 8 bands from spin, valley and layer. We use $\mu_a, \tau_a, s_a, a = 0, x, y, z$ to label the Pauli matrices in layer, valley and spin subspace respectively. We define $\psi_l(\mathbf{k}) = (c_{l;K;\uparrow}(\mathbf{k}), c_{l;K;\downarrow}(\mathbf{k}), c_{l;K';\uparrow}(\mathbf{k}), c_{l;K';\downarrow}(\mathbf{k}))^T$, where c is the annihilation operator of a hole and $l = t, b$ denotes the top layer and the bottom layer. We can then define $\psi(\mathbf{k}) = (\psi_t^T(\mathbf{k}), \psi_b^T(\mathbf{k}))^T$, where $\mathbf{k}$ is the absolute momentum relative to the same origin for both layers and both valleys. It is also convenient to define a relative momentum $\tilde{\mathbf{k}}_{l;\tau} = \mathbf{k} - \mathbf{K}_{l;\tau}$, where $\mathbf{K}_{l;\tau}$ is the momentum of the valley $\tau$ at layer $l$. In the following we use $\tilde{\mathbf{k}}$ for simplicity.

The Hamiltonian for the continuum model in the hole picture is then:

$$H = H_0 + H_M,$$
$$H_0 = \sum_{\mathbf{k}} \psi^\dagger(\mathbf{k}) \left( \frac{\hbar^2 \tilde{\mathbf{k}}^2}{2m^*} \mu_0 + \frac{D}{2} \mu_z \right) \otimes \tau_0 \otimes s_0 \psi(\mathbf{k})$$
$$+ \sum_{\mathbf{k}} \psi^\dagger(\mathbf{k}) \beta \mu_z \otimes \tau_z \otimes s_z \psi(\mathbf{k})$$
$$+ \sum_{\mathbf{k}} \psi^\dagger(\mathbf{k}) \mu_0 \otimes \tau_0 \otimes \alpha(D)(\tilde{k}_x s_y - \tilde{k}_y s_x) \psi(\mathbf{k})$$
$$+ \sum_{\mathbf{k}} \psi^\dagger(\mathbf{k}) \mu_z \otimes \tau_0 \otimes \alpha_-(\tilde{k}_x s_y - \tilde{k}_y s_x) \psi(\mathbf{k}),$$
$$H_M = -\sum_{\mathbf{k}} \sum_{j=1,3,5} \left( \psi^\dagger(\mathbf{k}) V e^{i\varphi} \mu_0 \otimes \tau_0 \otimes s_0 \psi(\mathbf{k}+\mathbf{G}_j) + \text{H.c} \right)$$
$$- \sum_{\mathbf{k}} \psi^\dagger(\mathbf{k}) w \mu_x \otimes \tau_0 \otimes s_0 \psi(\mathbf{k})$$
$$- \sum_{\mathbf{k}} \sum_{j=3,4} \left( \psi^\dagger(\mathbf{k}) w \mu_+ \otimes \tau_0 \otimes s_0 \psi(\mathbf{k}+\mathbf{G}_j) + \text{H.c} \right),$$
$$(1)$$

where $m^*$ is the effective hole mass, D is the voltage between the two layers, $\alpha(D)$ is the layer symmetric Rashba SOC induced by an external electric field, $\alpha_-$ is a layer-opposite Rashba SOC (LORSOC) allowed by symmetry (explained below) at D = 0, which has been studied in bilayer graphene[60], $\beta$ is the Ising spin-orbit coupling constant. $\mathbf{G}_i$s are the reciprocal vectors of the MBZ, and $w$ is an inter-layer tunneling strength parameter. We choose $\mathbf{G}_2$ to be $(-\frac{4\pi}{\sqrt{3}a_M}, 0)^T$, where $a_M \approx \frac{0.328}{\theta}$ nm ($\theta$ is the twist angle) is the moiré superlattice constant, and all the other G vectors are generated by 60-degree counterclockwise rotations of $\mathbf{G}_2$ (see Fig.2).

At $D = 0$, there is a $C_{2x}$ symmetry which acts as: $\psi(k_x, k_y) \rightarrow \mu_x \otimes \tau_0 \otimes (-is_x)\psi(k_x, -k_y)$. It enforces $\alpha(D) =$

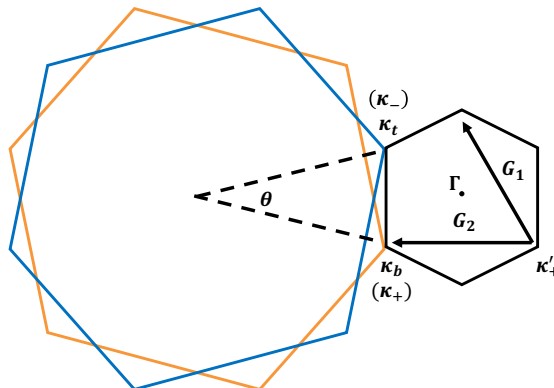

FIG. 2: The blue and orange hexagons are the Brillouin zones of the top and bottom layer in the hole picture, respectively. Twisting them with an angle $\theta$ creates the moiré Brillouin zone depicted by the black hexagon.

$-\alpha(-D)$, so there is only the layer-opposite Rashba $\alpha_-$ term at $D = 0$. Also the moiré Hamiltonian $H_M$ is fully constrained by the $C_{2x}$ symmetry and another $C_3$ rotation symmetry which acts as $\psi(\tilde{\mathbf{k}}) \rightarrow \psi(C_3\tilde{\mathbf{k}})$. $C_3$ is defined relative to the valleys K and K' of the corresponding layer. We note that the $C_{2x}$ symmetry guarantees that the moiré potentials have the same phase $\varphi$ in the two layers for the same $\mathbf{G_j}$. In contrast, for the AA stacking, the $C_{2y}$ symmetry requires the phase $\varphi$ to be opposite for the two layers at the same $\mathbf{G_j}$. This is crucial for the approximate SU(4) symmetry at $D = 0$ for the twisted AB stacked homo-bilayer.

The LORSOC $\alpha_-$ in Eq.1 is allowed at D = 0 by symmetry. It is not clear that how it can arise microscopically, but it should be quite small given that there is no vertical electric field between the two TMD layers at $D = 0$. Typical Rashba SOC in systems with strong mirror symmetry breaking is around 30 meV·Å [61]. We believe $\alpha_-$ in our system at $D = 0$ is significantly smaller. However, in the following we will use a $\alpha_-$ at the order of 10 meV ·Å to obtain an upper bound of inter-layer hybridization. The layer isotropic Rashba SOC $\alpha(D)$ should be proportional to $D$. Based on previous first-principle calculations in bilayer MXY[62], we use $\alpha(\text{meV} \cdot \text{Å}) = AD(\text{V})$ as an estimation, where $A = \frac{14\text{meV}\cdot\text{Å}\cdot\text{nm/V}}{0.7\text{nm}}$ is a proportionality constant and 0.7nm is the estimated inter-layer distance.

We compute the band structure from Eq.1 with $\theta = 4$ degrees, $m^* = 0.45m_0$[63], $V = 7.9$ meV, $\varphi = 142°$, $w = 18$ meV[64], $\beta = 230$ meV[65], $\alpha_- = 20$meV · Å. We plot the first two moiré bands in the valley K with several values of $D$ in Fig.3. Even when the LORSOC $\alpha_- =$ is as large as 20meV ·Å, we can see that the bands from the two layers are almost decoupled, indicating a very small inter-layer tunneling. We note that the $\alpha_-$ used here is overestimated. Thus, we expect the inter-layer hybridization to be even weaker in realistic system.

## III. LATTICE MODEL AND INTER-LAYER HOPPING WHEN $n \leq 1$

From the band structure calculation, we can see that the bands of the two layers are almost decoupled even with the Rashba SOC included. If we construct a lattice model for

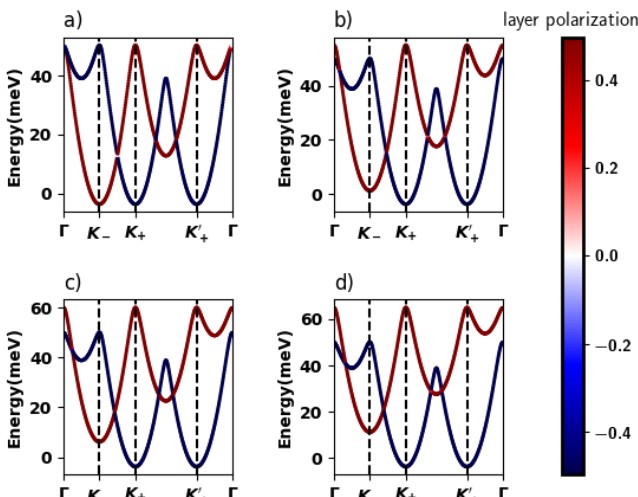

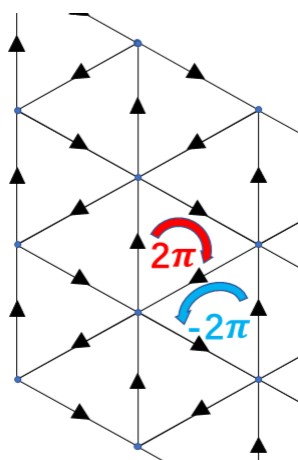

FIG. 4: Illustration of the hopping phase $\phi_{ij}^\tau$ for $\tau = K$ in Eq.2. We have $\phi_{ij}^K = -\phi_{ij}^{K'}$. Along the arrow we have $\phi_{ij} = \frac{2\pi}{3}$. Neighbouring triangles have opposite flux $\Phi = \pm 2\pi$.

FIG. 3: Dispersion relations of the first two moiré bands from the same valley $K$ at different displacement field $D$. The colors of the bands indicate layer polarization $P_z(\mathbf{k}) = \frac{1}{2}(c_t^\dagger(\mathbf{k})c_t(\mathbf{k}) - c_b^\dagger(\mathbf{k})c_b(\mathbf{k}))$. (a) D = 0 mV, $\alpha(D) = 0\text{meV} \cdot \text{Å}$ (b) D = 5 mV, $\alpha(D) = 0.1\text{meV} \cdot \text{Å}$ (c) D = 10 mV, $\alpha(D) = 0.2meV \cdot \text{Å}$ (d) D = 15 mV, $\alpha(D) = 0.3\text{meV} \cdot \text{Å}$. In all of the plots we set $\varphi = 142°$ and include a layer-opposite Rashba SOC $\alpha_- = 20\text{meV} \cdot \text{Å}$.

a band at one layer and one valley, it should be a simple tight-binding model on a triangular lattice. The question is how large the inter-layer tunneling is. To estimate it, we construct Wannier orbitals for the lowest two bands(from the two layers) at each valley on a triangular lattice. The two valleys are related by the time reversal symmetry, so we can only focus on one valley. We obtain an extended Hubbard model (see Appendix A. for details):

$$
\begin{aligned}
H = &-t \sum_{\langle ij \rangle} \sum_{\tau=K,K'} \left( (c_{ib\tau}^\dagger e^{i\phi_{ij}^\tau} c_{jb\tau} + c_{it\tau}^\dagger e^{-i\phi_{ij}^\tau} c_{jt\tau}) + \text{H.c} \right) \\
&+ t_z \sum_i \sum_\tau (c_{it\tau}^\dagger c_{ib\tau} + \text{H.c}) \\
&+ \frac{1}{2} \sum_i \sum_{l=t,b} U n_{il}(n_{il} - 1) + U' \sum_i n_{it} n_{ib} \\
&+ \sum_{\langle ij \rangle} \sum_l V n_{il} n_{jl} + \sum_{\langle ij \rangle} V'(n_{it} n_{jb} + n_{ib} n_{jt}),
\end{aligned}
\tag{2}
$$

where $\langle ij \rangle$ refers to the nearest neighbors, t is the real nearest-neighbour intra-layer hopping, $t_z$ is the on-site inter-layer tunneling. Because spin $S_z$ is not a good quantum number, we ignore the spin index. $S_z$ should be locked to valley, so one can view valley as the usual spin 1/2 in the familiar Hubbard model. $n_{i;l}$ is the density at layer $l = t, b$ (summed over valleys), $\phi_{ij}^K = -\phi_{ij}^{K'}$, $\phi_{ij}^\tau = \pm \frac{2\pi}{3}$ so that the flux through each triangle is $\pm 2\pi$ as shown in Fig.4. $U$ and $U'$ are the intra-layer and inter-layer on-site repulsion, $V$ and $V'$ are the intra-layer and inter-layer repulsion among nearest neighbours. We use $\mathbf{a}_1 = (0, a_M)^T$, $\mathbf{a}_2 = (\frac{\sqrt{3}}{2}a_M, -\frac{1}{2}a_M)^T$ to be the triangular lattice primitive vectors, $a_M$ is the moiré

superlattice constant. Note that we are free to choose the phase of $t_z$. In this paper, we let it be real.

We plot t in Fig.5 (a) and $\frac{t_z}{t}$ in Fig.5 (b) at D = 0 with several $\alpha_-$ as functions of the twist angle $\theta$. The on-site interlayer tunneling is smaller than 1% of the nearest neighbour intra-layer hopping at $D = 0$ for $\alpha_- \leq 5$ meV $\cdot$ Å and $\theta > 3°$. We can see that $\frac{t_z}{t}$ decreases with $\theta$, mainly because $t$ increases with the twist angle. We also want to check whether this result depends on the choice of $\varphi$ in our model. We provide a plot of $\frac{t_z}{t}$ as a function of $\varphi$ in Fig.6. One can see that the choice of $\varphi$ does not affect the order of magnitude of $\frac{t_z}{t}$.

### A. A more convenient gauge

It is convenient to gauge away the phase of the nearest neighbor hopping. Let us define $\Psi_i = (c_{itK}, c_{itK'}, c_{ibK}, c_{ibK'})^T$. The flux $\Phi = \pm 2\pi$ in each triangle can be gauged away by the transformation: $\Psi_i \rightarrow \Psi_i e^{i\mu_z \otimes \tau_z \kappa \cdot \mathbf{r}_i}$, $\kappa = (\frac{2\pi}{3a_M}, -\frac{2\pi}{\sqrt{3}a_M})$, $\kappa \cdot \mathbf{r}_i = \pm \frac{2\pi}{3}$. Note that $\kappa$ is a corner of the MBZ.

After the gauge transformation, we reach a new lattice model:

$$
\begin{aligned}
H = &-t \sum_{\langle ij \rangle} \sum_{\tau=K,K'} \sum_{l=t,b} (c_{il\tau}^\dagger c_{jl\tau} + H.c) \\
&+ t_z \sum_i (\Psi_{ib}^\dagger e^{2\kappa \cdot \mathbf{r}_i i \tau_z} \Psi_{it} + H.c) \\
&+ \frac{1}{2} \sum_i \sum_a U n_{il}(n_{il} - 1) + U' \sum_i n_{it} n_{ib} \\
&+ \sum_{\langle ij \rangle} \sum_l V n_{il} n_{jl} + \sum_{\langle ij \rangle} V'(n_{it} n_{jb} + n_{ib} n_{jt}),
\end{aligned}
\tag{3}
$$

where $\Psi_{il}(l = t, b) = (c_{ilK}, c_{ilK'})^T$.

If we ignore the $t_z$ term and set $U = U', V = V'$, the above model has a $U(4)$ symmetry. The cost of the gauge transformation is to add a position dependent phase on the inter-layer on-site hopping. In the following we will use this gauge so that the approximate SU(4) symmetry is explicit.

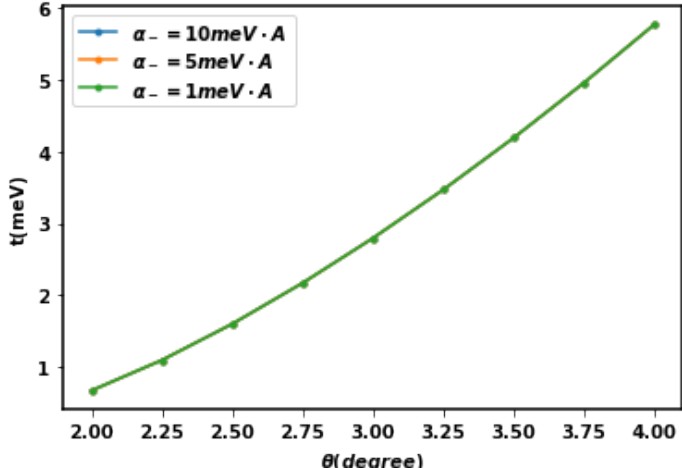

(a)

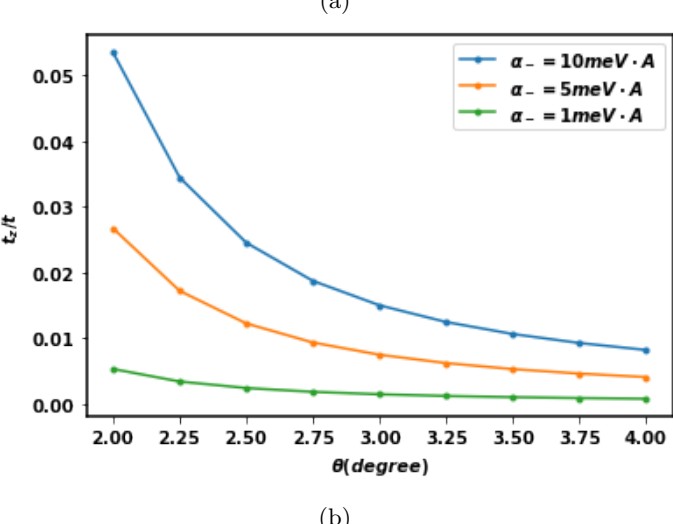

(b)

FIG. 5: (a)Nearest-neighbour intra-layer hopping $t$ vs the twist angle $\theta$ at D = 0, $\varphi = 142°$. t is independent of $\alpha_-$.
(b)Ratio of on-site inter-layer tunneling $t_z$ to nearest-neighbour intra-layer hopping $t$ vs the twist angle $\theta$ at D = 0 with different LORSOC constants $\alpha_-$. $\varphi = 142°$.

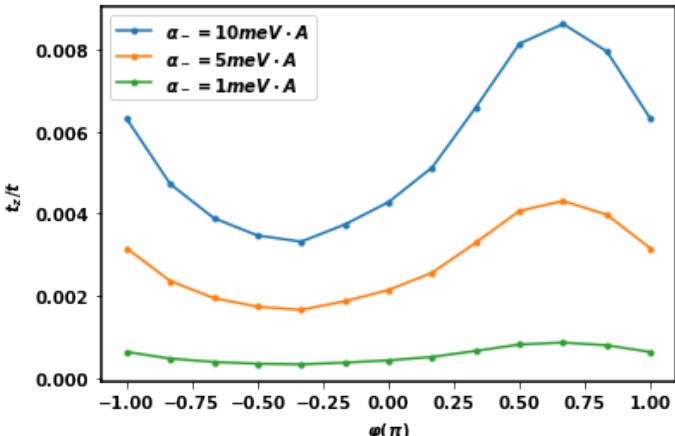

FIG. 6: Ratio of on-site inter-layer tunneling $t_z$ to nearest-neighbour intra-layer hopping $t$ vs $\varphi$ at D = 0, $\alpha_- = 1, 5, 10 \text{meV} \cdot \text{Å}$, $\theta = 4°$.

## IV. APPROXIMATE SU(4) SPIN MODEL AND THE EFFECT OF INTER-LAYER HOPPING AT $n = 1$

### A. Approximate SU(4) spin model

In a typical system, $\frac{t}{U}$ is about $0.01 \sim 0.1$[66], so we can obtain a spin model through a Schrieffer-Wolff transformation of Eq.3 in $t/U$ [67] for the Mott insulating phase at total filling $n = 1$. We define a layer pseudo-spin operator $\mathbf{P} = \frac{1}{2}\boldsymbol{\mu}$ and a spin-valley operator $\mathbf{S} = \frac{1}{2}\boldsymbol{\tau}$. The effective spin model is:

$$
\begin{aligned}
H_S =& \frac{J}{4} \sum_{\langle ij \rangle} (4\mathbf{P}_i \cdot \mathbf{P}_j + P_i^0 P_j^0)(4\mathbf{S}_i \cdot \mathbf{S}_j + S_i^0 S_j^0) \\
&+ \delta J \sum_{\langle ij \rangle} \left( (P_i^x P_j^x + P_i^y P_j^y)(4\mathbf{S}_i \cdot \mathbf{S}_j + S_i^0 S_j^0) \right) \\
&+ 2(\delta J + \delta V) \sum_{\langle ij \rangle} P_i^z P_j^z \\
&+ 2t_z \sum_i (P_i^x \cos 2\kappa \cdot \mathbf{r}_i - 2S_i^z P_i^y \sin 2\kappa \cdot \mathbf{r}_i),
\end{aligned}
\tag{4}
$$

where $J = \frac{2t^2}{U-V}, \delta J = \frac{2t^2}{U'-V'} - \frac{2t^2}{U-V}, \delta V = V - V'$.

$\delta J, \delta V$ are anisotropy terms breaking SU(4) spin symmetry due to the distance between the two layers. $\frac{\delta J}{J}, \frac{\delta V}{J}$ are estimated to be 0.2 and 0.3 at $\theta = 3°$[48]. When $\delta J = \delta V = t_z = 0$, the above model is an SU(4) Heisenberg model of which the ground state was shown to be in a plaquette order[48]. It is interesting to note that the inter-layer tunneling term $t_z$ acts as a sublattice-dependent transverse Ising field in the layer pseudo-spin space.

### B. Layer pseudo-spin magnetism and effect of transverse Ising field

Now, we study the effect of the sublattice dependent transverse Ising field. For simplicity, we apply a strong magnetic field to polarize the valleys in order to neglect $\mathbf{S}$ in Eq. 4. Then, we only need to consider an XXZ model with a transverse Ising field coupled to the layer pseudospin $\mathbf{P}$:

$$
\begin{aligned}
H =& J_{xy} \sum_{\langle ij \rangle} (P_i^x P_j^x + P_i^y P_j^y) + J_z \sum_{\langle ij \rangle} P_i^z P_j^z \\
&- \sum_i (H_x(\mathbf{r}_i) P_i^x - H_y(\mathbf{r}_i) P_i^y - D P_i^z), \\
&J_{xy} = 2(J + \delta J), \\
&J_z = 2(J + \delta J + \delta V), \\
&H_x = -2t_z \cos 2\kappa \cdot \mathbf{r}_i, \\
&H_y = 2t_z \sin 2\kappa \cdot \mathbf{r}_i,
\end{aligned}
\tag{5}
$$

One can see that the potential difference $D$ now plays the role of a Zeeman field in the layer pseudo-spin space. The inter-layer tunneling $t_z$ acts as a sublattice dependent transverse Ising field. Without the transverse Ising field, this is an XXZ model with a Zeeman field, which has been studied [68–70]. For example, it is known that there is a $2\langle P_z \rangle = \frac{1}{3}$

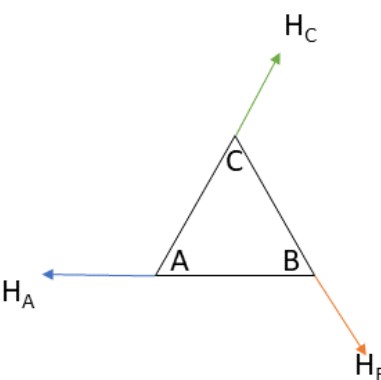

FIG. 7: Ilustration of the sublattice-dependent effective transverse Ising field from the inter-layer tunneling $t_z$ in the $P_x, P_y$ space. Here $A, B, C$ label three sublattices. $\mathbf{H}$ labels the direction of the transverse Ising field.

plateau in the magnetization when increasing the Zeeman field $D$. For our system, this plateau will be manifested in the layer polarization $P_z$. The effect of a uniform transverse Ising field has been studied[71]. But there is no discussion on the sublattice-dependent transverse Ising field in Eq. 5. We plot the directions of the field's projection in the x-y plane in Fig.7.

We compute the polarization curves of Eq. 5 numerically using the standard linear spin wave theory (see Appendix B for details). We use $\frac{J_{xy}}{J_z} = 0.1$ because $\frac{t^2}{U'} \sim O(0.01) - O(0.1), \delta V \sim O(1 meV)$. We show the polarization curves with the transverse Ising field $H_p = 2|t_z| = 0.1 J_z, 0.5 J_z$ in Fig. 8. The 1/3 plateau is clearly visible with $H_P = 0.1 J_z$. We believe that $H_p/J_z$ is smaller than 0.1 in typical systems because the inter-layer tunneling $t_z$ is found to be of O(0.01t) and $\frac{J_z}{t} \in O(1)$. Therefore, we propose to search for the 1/3 plateau in layer polarization by increasing the displacement field $D$ under a strong magnetic field in experiments[50].

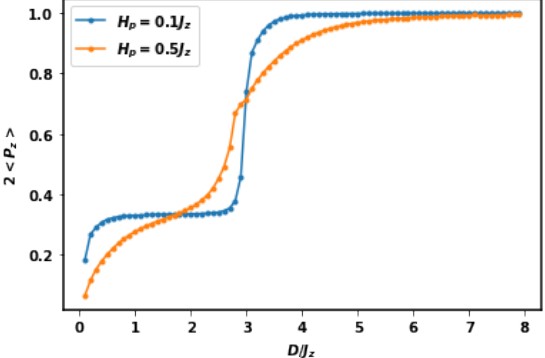

FIG. 8: Layer polarization $2\langle P_z \rangle$ vs displacement field $\frac{D}{J_z}$ at different sublattice dependent transverse Ising field $H_p = 2t_z$. We fix $\frac{J_{xy}}{J_z} = 0.1$.

Then, we study the phase diagram of the above model by varying $D/J_z$ and $H_p/J_z$ while fixing $J_{xy}/J_z = 0.1$. The result is shown in Fig. 9 (see Appendix.B for details). We find that the 1/3 plateau survives if $H_p/J_z < 0.4$. When $H_p$ is weak, the phases are similar to an XXZ model in a z-direction magnetic field[70], but an umbrella phase replaces the co-planar V-shape phase. When $H_p$ is strong, the $\vec{P}$ vectors form a 120-degree structure in the x-y plane. When a displacement field $D$ is applied, the $\vec{P}$ vectors form an umbrella structure. In usual twisted bilayer TMDs, we only realize the bottom part of the phase diagram because $\frac{H_p}{J_z}$ is small.

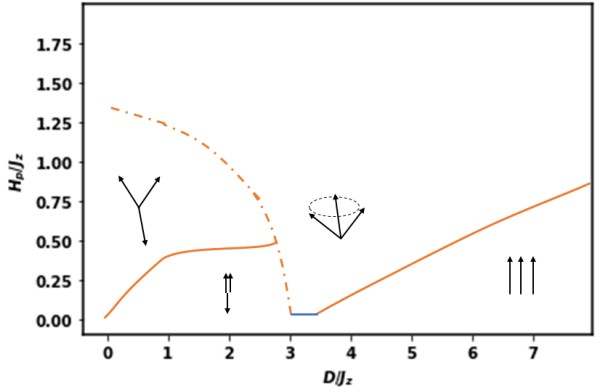

FIG. 9: Phase diagram of Eq.5 when $\frac{J_{xy}}{J_z} = 0.1$. $H_p$ is the effective sublattice dependent x-y plane magnetic field originating from the inter-layer hopping $t_z$. D is the voltage between two layers and is an effective magnetic field in the z-direction. The dashed lines mark first-order phase transitions where $\langle P_z \rangle$ jumps and solid lines mark second-order phase transitions. The $H_p = 0$ line was studied in [70]. Note that the umbrella structure becomes co-planar at $H_p = 0$ on the blue line. In experiments of the twisted AB stacked TMD homo-bilayer[50], one only has access to the bottom part where $\frac{H_p}{J_z}$ is small.

## V. HUBBARD MODEL ON A HONEYCOMB LATTICE WHEN $n > 1$

The single orbital Hubbard model in the above is valid only when $n \leq 1$. When $n > 1$, the additional hole may want to enter another orbital in a different position inside the moiré unit cell to avoid the large Hubbard U due to the double occupancy[25]. To capture this effect, we need to keep at least another orbital from a higher band. As we have demonstrated that the inter-layer tunneling is quite small, in the following we ignore the Rashba SOC in Eq. 1. It is also convenient to shift the valence band maximum of the two layers to the $\Gamma$ point of the MBZ for each valley. This is equivalent to the gauge transformation we did to reach Eq.3 in the previous section. Then we have two degenerate bands from the two layers at the same valley. In together we have four fold degeneracy for each moiré band coming from the combination of layer and valley.

In the hole picture, we use the two lowest bands (the

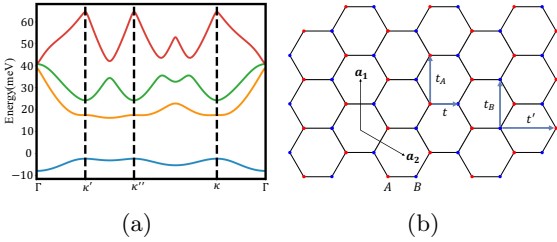

FIG. 10: (a) The first four energy bands when $\theta = 2°$ and $\alpha_{-}0$ in AB stacking case. (b) Illustration of a honeycomb lattice. The red dots and the blue dots represent sublattice A and B respectively.

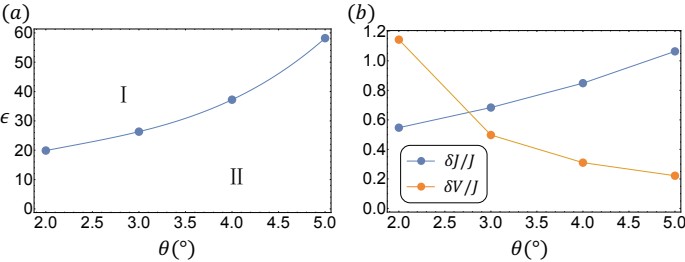

FIG. 11: (a) The phase diagram of the four-flavor Hubbard model when $n = 1 + x$. The two regions: (I) The additional holes still prefer to stay on sublattice B. (II) The additional holes go to sublattice A. (b) Dependence of $\frac{\delta J}{J}$ and $\frac{\delta V}{J}$ on the twist angle $\theta$. Here, we use $\epsilon = 10$ as an example.

blue and yellow bands in Fig.10(a)) to build a lattice model. It turns out that the Wannier orbitals[72] of the these two bands are on the two sublattices of a honeycomb lattice (see Appendix C. for details). This leads to a four-flavor Hubbard model:

$$
\begin{aligned}
H = & -\sum_{\substack{\langle ij \rangle \\ \mu,\tau}} (t c_{i,\mu,\tau}^\dagger c_{j,\mu,\tau} + \text{H.c.}) - \sum_{\substack{\langle\langle\langle ij \rangle\rangle\rangle \\ \mu,\tau}} (t' c_{i,\mu,\tau}^\dagger c_{j,\mu,\tau} + \text{H.c.}) \\
& - \sum_a \left( \sum_{\substack{\langle\langle i_a j_a \rangle\rangle \\ \mu,\tau}} (t_a c_{i_a,\mu,\tau}^\dagger c_{j_a,\mu,\tau} + \text{H.c.}) \right) \\
& + \frac{\Delta}{2} \left( \sum_{i_A} n_{i_A} - \sum_{i_B} n_{i_B} \right) \\
& + \sum_a \sum_{i_a,\mu} \frac{U_a}{2} n_{i_a,\mu}(n_{i_a,\mu} - 1) + \sum_a \sum_{i_a} U_a' n_{i_a,t} n_{i_a,b} \\
& + \sum_{\langle ij \rangle, \mu} V n_{i,\mu} n_{j,\mu} + \sum_{\langle ij \rangle} (V' n_{i,t} n_{j,b} + \text{H.c.}),
\end{aligned}
\tag{6}
$$

where $c_{i,\mu,s}^\dagger$ is the creation operator of localized Wannier orbitals (see Appendix C). Here $\mu = t, b$ labels the layer and $\tau = K, K'$ labels the valley. $\langle \ldots \rangle$, $\langle\langle \ldots \rangle\rangle$, $\langle\langle\langle \ldots \rangle\rangle\rangle$ represents nearest neighbor, next nearest neighbor, next next nearest neighbor, respectively. $i_a(a = A, B)$ represents the site on A,B sublattice, which is defined in Fig.10(b). The interaction parameters $U_a$, $U_a'$, $V$, $V'$ are calculated via projecting the Coulomb repulsion $\tilde{U}(\mathbf{r}) = e^2/\epsilon r$ onto the Wannier orbitals, the details are in Appendix C.

The values of the parameters are listed in Table I for a few twist angles. Because of the sublattice potential $\Delta$ term, holes occupy the B sublattice when $n \leq 1$, leading to a single orbital Hubbard model on the triangular lattice formed by B only, which reduces to the model used in the previous section. However, when $n > 1$, the additional holes prefer to enter A if $U_B' > V' + \Delta$. This happens when the dielectric constant $\epsilon$ is smaller than a threshold. The boundary between the two regions is plot in Fig. 11(a). In the following we focus on the region II. There is already experimental evidence that the region II is realized in the real experiment[50].

Now we have a Mott insulator on a honeycomb lattice at the total filling $n = 2$. The resulting spin model in the large $U/t$ limit is:

$$
\begin{aligned}
H_S = & \frac{J}{4} \sum_{\langle ij \rangle} (4\mathbf{P}_i \cdot \mathbf{P}_j + P_i^0 P_j^0)(4\mathbf{S}_i \cdot \mathbf{S}_j + S_j^0 S_j^0) \\
& + \delta J \sum_{\langle ij \rangle} (P_i^x P_j^x + P_i^y P_j^y)(4\mathbf{S}_i \cdot \mathbf{S}_j + S_i^0 S_j^0) \\
& + 2(\delta J + \delta V) \sum_{\langle ij \rangle} P_i^z P_j^z,
\end{aligned}
\tag{7}
$$

where $\langle ij \rangle$ stands for nearest neighbor AB bond of the honeycomb lattice. $J = \frac{t^2}{U_A - V + \Delta} + \frac{t^2}{U_B - V - \Delta}$, $\delta J = \frac{t^2}{U_A' - V' + \Delta} + \frac{t^2}{U_B' - V' - \Delta} - \frac{t^2}{U_A - V + \Delta} - \frac{t^2}{U_B - V - \Delta}$, $\delta V = V - V'$. $\frac{\delta J}{J}$ and $\frac{\delta V}{J}$ are plotted as functions of the twist angle in Fig.11. We only keep nearest-neighbor hopping because $t^2$ is usually larger enough than $t'^2, t_A^2, t_B^2$, which can be seen from Table I. Strictly speaking there should still be an inter-layer hopping term similar to the $t_z$ term in Eq.3. We will ignore it because it is very small as discussed in Sec. III.

If we set $\delta J = \delta V = 0$, the above model is a SU(4) Heisenberg model on honeycomb lattice. Previous works have suggested that the ground state is a Dirac spin liquid with $\pi$ flux ansatz[56–58] with $N_f = 2 \times 4 = 8$ Dirac fermions. Here 4 is from the four flavors and each flavor hosts two Dirac fermions. If $\delta V/J$ is large enough, the ground state must have $P_z = \frac{1}{2}$ on one sublattice and $P_z = -\frac{1}{2}$ on the other sublattice. It is interesting to study the phase transition between the Dirac spin liquid and this layer pseudo-spin density wave state through tuning $\delta J, \delta V$. The monopole operator[59] in the Dirac spin liquid may also play an important role in the potential phase transitions. We leave to future work to carefully study the phase diagram and possible quantum criticalities in Eq. 7.

## VI. CONCLUSION

In this paper, we derive a lattice model for the twisted AB stacked TMD homo-bilayer through continuum model calculation and Wannier orbital construction. Without Rashba SOC, we have an approximate SU(4) Hubbard model. We consider the effect of the Rashba SOC and find that it generates a small inter-layer tunneling, meaning that the violation of the SU(4) symmetry due to inter-layer tunneling is small. At total filling $n = 1$, we derive an approximate SU(4) spin

| $\theta$ | $\Delta$ | $t$ | $t'$ | $t_A$ | $t_B$ | $\epsilon U_A$ | $\epsilon U_B$ | $\epsilon U'_A$ | $\epsilon U'_B$ | $\epsilon V$ | $\epsilon V'$ |
|---|---|---|---|---|---|---|---|---|---|---|---|
| 2 | 19.15 | 5.26 | 0.788 | -0.764 | -0.587 | 479.22 | 825.44 | 405.27 | 640.10 | 276.13 | 259.27 |
| 3 | 14.25 | 12.58 | 2.926 | -1.328 | -1.945 | 694.20 | 1005.68 | 544.28 | 729.95 | 389.02 | 354.83 |
| 4 | 9.86 | 22.27 | 6.513 | -2.205 | -3.866 | 873.12 | 1174.04 | 639.25 | 798.28 | 489.45 | 431.29 |
| 5 | 6.18 | 34.36 | 11.38 | -3.466 | -6.216 | 1046.88 | 1342.77 | 717.40 | 856.04 | 583.76 | 496.37 |

TABLE I: Dependence of the kinetic and interaction parameters on different twist angle $\theta$. The unit of $\theta$ is $^\circ$ and that of energy is meV. We assume that the distance between the top and the bottom layer is 0.7nm.

model which reduces to an XXZ model for the layer pseudo-spin with an additional sublattice-dependent transverse Ising field in a strong magnetic field. We study the phase diagram and find a 1/3 plateau in the layer polarization curve when increasing the displacement field $D$. When $n = 2$, the two holes prefer to stay in the A, B sublattices of a honeycomb lattice in a certain regime. This results in an approximate SU(4) spin model on a honeycomb lattice, which may host a Dirac spin liquid ground state. The lattice models derived in this work offer a starting point to explore the various strongly correlated physics in twisted AB-stacked TMD homo-bilayer.

**ACKNOWLEDGMENTS**

We thank Junyi Zhang for helpful discussions. This work is supported by a startup fund from the Johns Hopkins University. The numerical simulation was carried out at the Advanced Research Computing at Hopkins (ARCH) core facility (rockfish.jhu.edu), which is supported by the National Science Foundation (NSF) grant number OAC 1920103.

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

## Appendix A: Construction of the tight-binding model

We want to build a two-layer tight-binding model of the lowest moiré band which is four-fold degenerate and consists of two valleys and two layers. As the two valleys are decoupled and are related by the time reversal symmetry, we only focus on the valley K. We start from the bands of Eq.1 which can be obtained from numerical diagonalization. We choose

the lowest two bands from the two layers at valley K:

$$H_{TB}(\mathbf{k}) = -\sum_{a=1,2}\sum_{b=1,2} t_{ab}c_a^\dagger(\mathbf{k})c_b(\mathbf{k}). \quad (A1)$$

where $a, b = 1, 2$ are band indices. If there is no Rashba SOC, these two bands are from the two layers and are decoupled. With a finite Rashba SOC, the band indices are not layer indices anymore. To build a two-layer model, we define a layer polarization operator $P_z(\mathbf{k}) = \frac{1}{2}(c_t^\dagger(\mathbf{k})c_t(\mathbf{k}) - c_b^\dagger(\mathbf{k})c_b(\mathbf{k}))$ and project it into the lowest two bands. We transform to a new basis where $P_z(\mathbf{k})$ is diagonal at every momentum $\mathbf{k}$. Then in this new basis we reach a two-band Hamiltonian:

$$H_{TB}(\mathbf{k}) = -\sum_{l=t,b}\sum_{l'=t,b} t_{ll'}c_l^\dagger(\mathbf{k})c_{l'}(\mathbf{k}). \quad (A2)$$

Note that $t_{bb}(\mathbf{k}), t_{tt}(\mathbf{k})$ are invariant under a change of phase $c_t(k) \rightarrow c_t(k)e^{i\theta_t(k)}, c_b(k) \rightarrow c_b(k)e^{i\theta_b(k)}$, but $t_{bt}(k), t_{tb}(k)$ are not. We fix the relative phase $\theta_t(\mathbf{k}) - \theta_b(\mathbf{k})$ by enforcing that $P_x(\mathbf{k}) \approx \frac{1}{2}\sigma_x$. $P_x(\mathbf{k})$ is obtained by projecting $\frac{1}{2}(c_t^\dagger(\mathbf{k})c_b(\mathbf{k}) + c_b^\dagger(\mathbf{k})c_t(\mathbf{k}))$ into the lowest two bands, similar to $P_z(\mathbf{k})$. Fig.12 shows that our gauge is smooth and we only need to consider $t_z(0,0)$.

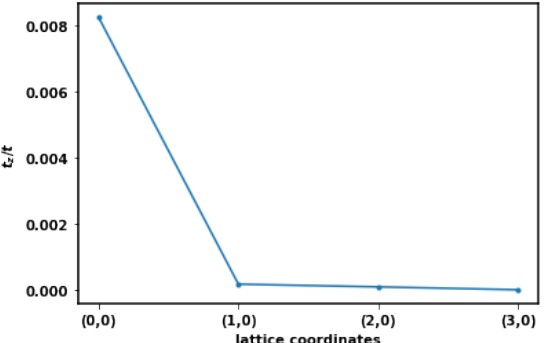

FIG. 12: $\frac{t_z(\mathbf{r})}{t}$ at $D = 5mV, \alpha = 0.1meV \cdot \text{Å}, \alpha_- = 10meV \cdot \text{Å}$ as a function of lattice coordinates at twist angle $\theta = 4°$. $t_z(0,0)$ is clearly dominant.

Consider the term $-\sum_{\mathbf{k}} t_{bt}(\mathbf{k})c_b^\dagger(\mathbf{k})c_t(\mathbf{k})$. We plug in the Wannier functions $c(\mathbf{k}) = \frac{1}{\sqrt{N}}\sum_{\mathbf{r}}\exp(i\mathbf{k}\cdot\mathbf{r})c(\mathbf{r})$ and get

$-\frac{1}{N}\sum_{\mathbf{k}}\sum_{\mathbf{r}}\sum_{\mathbf{r}'} t_{bt}(\mathbf{k})\exp[-i\mathbf{k}\cdot(\mathbf{r}-\mathbf{r}')]c_b^\dagger(\mathbf{r})c_t(\mathbf{r}')$. Let $\mathbf{r} - \mathbf{r}' = m\mathbf{a}_1 + n\mathbf{a}_2, t_{bt}(m,n) = -\frac{1}{N}\sum_{\mathbf{k}} t_{bt}(\mathbf{k})\exp[-i\mathbf{k}\cdot(\mathbf{r}-\mathbf{r}')] = \sum_{\mathbf{k}} t_{bt}(\mathbf{k})\exp[-i\mathbf{k}\cdot(m\mathbf{a}_1+n\mathbf{a}_2)]$, where $\mathbf{a}_1, \mathbf{a}_2$ are two Bravais lattice vectors of the triangular lattice. The top-to-bottom-layer hopping term in the tight-binding model can now be written as $\sum_{\mathbf{r}}\sum_m\sum_n t_{bt}(m,n)c_b^\dagger(\mathbf{r} + m\mathbf{a}_1 + n\mathbf{a}_2)c_t(\mathbf{r})$. We only keep on-site and nearest-neighbour terms.

## Appendix B: Spin-Wave Theory

We want to study the quantum phase diagram of Eq. 5 with respect to the effective magnetic field. It is well-known that on a triangular lattice, the XXZ model has a three-sublattice magnetic structure[68–70]. We assume that the

basic magnetic structure is similar here. We choose the order parameter to be the average sublattice polarization which we can use spin wave theory[73, 74] to compute.

The main idea of spin wave theory is viewing the expectation of the pseudo-spin $\mathbf{P}$ as a classical vector and using three bosonic fields a,b,c for the three sublattices of a triangular lattice to do a Holstein-Primakoff (H-P) transformation such that $P_u^z = p - u^\dagger u, P_u^+ = \sqrt{2p}u, P_u^- = \sqrt{2p}u^\dagger$, u = a, b, c, and p is the magnitude of $\mathbf{P}$, where we have set $\hbar = 1$, taken the large-p limit and only kept the leading order term. This is called linear spin wave theory. In this approximation, the magnons are free. In the H-P transformation, we only keep terms of order $p^2$ and p and quadratic in the a,b,c fields. We then use the Fourier series $u(\mathbf{r}) = \frac{1}{\sqrt{N}}\sum_{\mathbf{k}} u(\mathbf{q})e^{i\mathbf{q}\cdot\mathbf{r}}, u = a, b, c$ to find the Hamiltonian in the momentum space. Note that $\mathbf{q}$ is in the Brillouin zone of only one of the sublattices. In the end, we set p = $\frac{1}{2}$ as an approximation.

In each sublattice, the P vector is in a different direction. Therefore, in Eq.5 it is convenient to transform the P vectors to their local frames.

### 1. Rotation of frame

Let the three sublattices be a,b,c. Firstly, we can rotate the x-y plane so that the direction of the magnetic field's projection in the x-y plane becomes the new x'-axis:
$(P_i^x, P_i^y)^T = \begin{bmatrix} \cos\omega_i & \sin\omega_i \\ -\sin\omega_i & \cos\omega_i \end{bmatrix} (P_i^{x'}, P_i^{y'})^T, i = a, b, c, \omega_i = \pi - 2\kappa\cdot\mathbf{r}_i$. The problem is now only in the x'-z plane. Then, we use spherical coordinates $(\theta, \phi)$ to parameterize the p vectors.

For convenience, we remove all the primes of the P vectors in the local frame in Eq. B1 below. Please keep in mind that all the axis labels below belong to the local frames. After the rotations, Eq. 5 becomes

$$H = 3(H_{ab} + H_{bc} + H_{ac}) - \sum_{i=a,b,c} H_x P_i^x \cos\theta_i$$
$$- H_z \sum_i P_i^z \cos\theta_i, \quad (B1)$$

where

$$H_{ab} = J_z[\alpha_{ab}P_a^z P_b^z + \lambda_{ab}P_a^x P_b^x + \zeta_{ab}P_a^y P_b^y + \mu_{ab}P_a^x P_b^y + \nu_{ab}P_b^x P_a^y)], \quad (B2)$$

, $H_{bc}, H_{ac}$ are defined similarly, $H_x = 2$. We have defined

$$\Delta = \frac{J}{J_z},$$
$$\epsilon_{ab} = \cos\omega_a \cos\omega_b + \sin\omega_a \sin\omega_b,$$
$$\eta_{ab} = \cos\omega_a \sin\omega_b - \sin\omega_a \cos\omega_b,$$
$$\alpha_{ab} = \cos\theta_a \cos\theta_b + \Delta\epsilon_{ab}\sin\theta_a \sin\theta_b,$$
$$\beta_{ab} = \sin\theta_a \sin\theta_b + \Delta\epsilon_{ab}\cos\theta_a \cos\theta_b,$$
$$\lambda_{ab} = \beta_{ab}\cos\phi_a \cos\phi_b + \Delta\epsilon_{ab}\sin\phi_a \sin\phi_b$$
$$+ \Delta\eta_{ab}(\cos\theta_b \sin\phi_a \cos\phi_b - \cos\theta_a \cos\phi_a \sin\phi_b),$$
$$\zeta_{ab} = \Delta\epsilon_{ab}\cos\phi_a \cos\phi_b + \beta_{ab}\sin\phi_a \sin\phi_b$$
$$+ \Delta\eta_{ab}(\cos\theta_a \sin\phi_a \cos\phi_b - \cos\theta_b \cos\phi_a \sin\phi_b),$$
$$\mu_{ab} = \beta_{ab}\cos\phi_a \sin\phi_b - \Delta\epsilon_{ab}\sin\phi_a \cos\phi_b$$
$$+ \Delta\eta_{ab}(\cos\theta_a \cos\phi_a \cos\phi_b + \cos\theta_b \sin\phi_a \sin\phi_b),$$
$$\nu_{ab} = \beta_{ab}\sin\phi_a \cos\phi_b - \Delta\epsilon_{ab}\cos\phi_a \sin\phi_b$$
$$- \Delta\eta_{ab}(\cos\theta_a \sin\phi_a \sin\phi_b + \cos\theta_b \cos\phi_a \cos\phi_b) \quad (B3)$$

. We have discarded terms that do not contribute to the leading-order ground state fluctuations such as $P^x P^z$. We choose $\mathbf{r}_a = (0, 0), \mathbf{r}_b = (a_M, 0), \mathbf{r}_c = (\frac{a_M}{2}, \frac{\sqrt{3}a_M}{2}) \implies \omega_a = \pi\omega_b = -\frac{\pi}{3}, \omega_c = \frac{\pi}{3}$. Note that equation B1 has an a,b,c permutation symmetry, so our choice of a,b,c in this paper is without loss of generality.

### 2. Hamiltonian

After the rotations above, H-P transformation, and Fourier transform, we get

$$H_{sw} = H_{cl} + 3p\sum_{\mathbf{k}} d^\dagger(\mathbf{k})M(\mathbf{k}, \frac{J_{xy}}{J_z}, \mathbf{H})d(\mathbf{k}), \quad (B4)$$

where $H_{cl}$ is the classical Hamiltonian

$$H_{cl} = 3p^2 J_z \sum_{i,j}[\Delta\epsilon_{ij}(\sin\theta_i \sin\theta_j \cos\phi_i \cos\phi_j + \sin\theta_i \sin\theta_j \sin\phi_i \sin\phi_j) + \cos\theta_i \cos\theta_j + \Delta\eta_{ij}(\sin\theta_i \sin\theta_j \cos\phi_i \sin\phi_j - sin\theta_i \sin\theta_j \sin\phi_i \cos\phi_j)]$$
$$- p\sum_i H_x \sin\theta_i \cos\phi_i - pH_z \sum_i \cos\theta_i, \quad (B5)$$

$\theta_a, \theta_b, \theta_c, \phi_a, \phi_b, \phi_c$ are spherical coordinates and minimize $H_{cl}$(We use scipy.optimize to perform the optimization.), $\mathbf{d}(\mathbf{k}) = (a_{\mathbf{k}}, b_{\mathbf{k}}, c_{\mathbf{k}}, a_{-\mathbf{k}}^\dagger, b_{-\mathbf{k}}^\dagger, c_{-\mathbf{k}}^\dagger)^T$, and $M = \begin{bmatrix} A & B \\ B & A \end{bmatrix}$,

$$A = \begin{bmatrix} -3(\alpha_{ab} + \alpha_{ac}) + h(\theta_a) & \frac{f}{2}[\lambda_{ab} + \zeta_{ab} + (\nu_{ab} - \mu_{ab})i] & \frac{f^*}{2}[\lambda_{ac} + \zeta_{ac} + (\nu_{ac} - \mu_{ac})i] \\ \frac{f^*}{2}[\lambda_{ab} + \zeta_{ab} - (\nu_{ab} - \mu_{ab})i] & -3(\alpha_{ab} + \alpha_{bc}) + h(\theta_b) & \frac{f}{2}[\lambda_{bc} + \zeta_{bc} + (\nu_{bc} - \mu_{bc})i] \\ \frac{f}{2}[\lambda_{ac} + \zeta_{ac} - (\nu_{ac} - \mu_{ac})i] & \frac{f^*}{2}[\lambda_{bc} + \zeta_{bc} - (\nu_{bc} - \mu_{bc})i] & -3(\alpha_{ac} + \alpha_{bc}) + h(\theta_c) \end{bmatrix},$$

$$B = \begin{bmatrix} 0 & \frac{f}{2}[\lambda_{ab} - \zeta_{ab} + (\nu_{ab} + \mu_{ab})i] & \frac{f^*}{2}[\lambda_{ac} - \zeta_{ac} + (\nu_{ac} + \mu_{ac})i] \\ \frac{f^*}{2}[\lambda_{ab} - \zeta_{ab} - (\nu_{ab} + \mu_{ab})i] & 0 & \frac{f}{2}[\lambda_{bc} - \zeta_{bc} + (\nu_{bc} + \mu_{bc})i] \\ \frac{f}{2}[\lambda_{ac} - \zeta_{ac} - (\nu_{ac} + \mu_{ac})i] & \frac{f^*}{2}[\lambda_{bc} - \zeta_{bc} - (\nu_{bc} + \mu_{bc})i] & 0 \end{bmatrix},$$

$$f(\mathbf{k}) = e^{i\mathbf{k}\cdot\delta_1} + e^{i\mathbf{k}\cdot\delta_2} + e^{i\mathbf{k}\cdot\delta_3}, h(\theta) = \frac{H_z}{p}\cos\theta + \frac{H_x}{p}\sin\theta, \delta_1 = (1,0), \delta_2 = (-\tfrac{1}{2}, \tfrac{\sqrt{3}}{2}), \delta_3 = (-\tfrac{1}{2}, -\tfrac{\sqrt{3}}{2}).$$

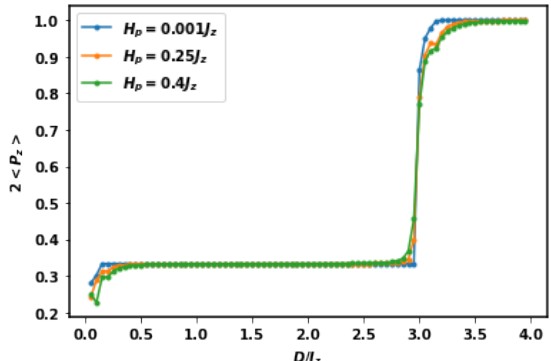

FIG. 13: Layer polarization $2\langle P_z \rangle$ vs displacement field $\frac{D}{J_z}$ at different sublattice dependent transverse Ising field $H_p = 2t_z$ near phase boundaries. $\frac{J_{xy}}{J_z} = 0.1$.

Note that $\lambda_{ij} + \zeta_{ij}$ and $\nu_{ij} - \mu_{ij}$ can be simplified with angle sum and difference identities.

### 3. Sublattice polarization

Our goal is to compute the average sublattice polarization. The sublattice polarization of a, for example, is given by the ground state expectation of $a^\dagger a$, which can be obtained from the Bogoliubov transformation that diagonalizes M. We follow [75] to obtain the Bogoliubov transformation matrix U such that $\mathbf{d}(\mathbf{k}) = U\gamma(\mathbf{k})$, where $\gamma(\mathbf{k})$ is a diagonal basis for M. Once we have U, the layer pseudo-spin reduction due to quantum fluctuation for sublattice l is

$$\Delta p_l = \frac{1}{N} \sum_{\mathbf{k}} |U_{i4}|^2 + |U_{i5}|^2 + |U_{i6}|^2 \qquad (B6)$$

, i = 1 if l = a, i = 2 if l = b, and i = 3 if l = c[74]. Finally, the average sublattice layer polarization in the z-direction is

$$\frac{p(\cos\theta_a + \cos\theta_b + \cos\theta_c)}{3} - \sum_l \frac{\Delta p_l \cos\theta_l}{3} \qquad (B7)$$

### 4. Phase diagram

We explain how we plot Fig.9 here. First, we set $H_p = 0.001J_z$ to study the model's behavior near the $H_p = 0$ limit.

As Fig.13 shows, the transition between the Y-shape phase and the $\frac{1}{3}$ plateau happens near $D = 0.3J_z$ and the transition between the $\frac{1}{3}$ plateau and the umbrella phase happens near $D = 3J_z$. The optimized $\theta_a, \theta_b, \theta_c < \frac{\pi}{2}$, confirming the umbrella phase. Then around $D = 3.3J_z$, the system is saturated. Then we use $H_p = 0.1J_z$ to extend the boundaries. All the boundaries are determined similarly with curves similar to the ones in Fig.9 and Fig.13. We find that the Y-shape phase disappears between $H_p = 1.3J_z$ and $H_p = 1.35J_z$.

The layer polarization clearly jumps at first-order phase transitions but not at second-order ones.

### Appendix C: Construction of the honeycomb lattice model

In the main text, we constructed an effective lattice model to describe the first 2 energy bands (each is four fold degenerate coming from the spin and the layer). The appropriate Wannier orbitals for the construction are[47]:

$$c_n^\dagger(\mathbf{x}_0) = \frac{1}{\sqrt{N}} \sum_{\mathbf{k},a} e^{-i\mathbf{k}\cdot\mathbf{x}_0} c_a^\dagger(\mathbf{k}) U_{an}(\mathbf{k}), \qquad (C1)$$

where $U_{an}(\mathbf{k})$ is an $m \times m$ unitary matrix, $c_a^\dagger(\mathbf{k})$ is the creation operator of the $a^{th}$ band's Bloch wave function.

$U_{an}(\mathbf{k})$ can be determined by projecting the original Bloch orbitals to well-localized wave functions[72]. We first calculate $A_{an}(\mathbf{k}) = \langle \psi_a(\mathbf{k}) | g_n(\mathbf{k}) \rangle$. Here, $\psi_a(\mathbf{k})$ and $g_n(\mathbf{k})$ are the Bloch state and the trial state respectively. Then, $U = A(A^\dagger A)^{-1/2}$. As for the trial wave function, we choose it to be:

$$g_n(\mathbf{x}) = \frac{1}{\pi^{1/2}\alpha} e^{-\frac{(\mathbf{x}-\mathbf{x}_n)^2}{2\alpha^2}}, \qquad (C2)$$

where $\alpha = a_M/6$ and $\mathbf{x}_n$ are determined by the center of the original $n^{th}$ Bloch state.

If we consider the case with Rashba coupling $\alpha_- = 0$ and include first two bands, i.e., $m = 2$, the centers of Wannier orbitals form a honeycomb lattice as required by the $C_3$ symmetry. Their positions correspond to the center of the original Bloch states, which can be obtained by calculating the eigenvalue of the $C_3$ operator. We observed that the $C_3$ eigenvalues of the first band are $1, e^{-i\frac{2\pi}{3}}, e^{i\frac{2\pi}{3}}$ at $\Gamma, \kappa_+, \kappa_-$. For the second band, they are $1, e^{i\frac{2\pi}{3}}, e^{-i\frac{2\pi}{3}}$ respectively. This constrains the Wannier orbitals of the two bands to be in different positions within a moiré unit cell.

We find that the kinetic and the interaction parameters in Eq.6 can be calculated by projecting the energy band and the Coulomb interaction onto the new basis. We take $t$ and $\epsilon U_A$ as an example, suppose that the wannier function localized at $\mathbf{x}_0$ in sublattice $n$ is $\phi_n(\mathbf{x} - \mathbf{x_0})$, then $t$ and $\epsilon U_A$ are:

$$t = \frac{1}{N} \sum_{\mathbf{k},a} U_{Aa}^{\dagger}(\mathbf{k}) \xi_a(\mathbf{k}) U_{aB}(\mathbf{k}) e^{-i(\mathbf{k}\cdot\mathbf{e_x})\frac{a_M}{\sqrt{3}}},$$

$$\epsilon U_A = \int d^3\mathbf{x} d^3\mathbf{x}' \phi_A^*(\mathbf{x}') \phi_A^*(\mathbf{x}) \frac{e^2}{|\mathbf{x} - \mathbf{x}'|} \phi_A(\mathbf{x}) \phi_A(\mathbf{x}'),$$

(C3)

where $\xi_a(\mathbf{k})$ is the dispersion relation of the $a^{th}$ band. In reality we first express the interaction in the momentum space with form factors and then do the Fourier transformation, following the same procedure as in Ref. 47.