# Peer review of "Approximate SU(4) spin models on triangular and honeycomb lattices in twisted AB-Stacked WSe$_2$ homo-bilayer"

_SciPost Physics_

## Round 1 · Referee Report · Antonio Manesco (Referee 1) · 2023-5-20

Strengths

1 - The work is timely, considering that there are recent experiments performed on these platforms.
2 - The authors found that at filling factor n=2 the system behaves as a SU(4)-symetric honeycomb lattice.

Weaknesses

1 - The code used in the simulations is not publicly available.
2 - The consideration of symmetry-breaking terms in their analysis is unclear since it seems to be irrelevant under experimental conditions.

Report

The authors study twisted AB homobilayers of transition metal dichalcogenides. They consider a previously neglected Rashba spin-orbit coupling effect in the interlayer coupling. Because of Rashba spin-orbit coupling effects, the authors show that the system effectively behaves as an approximate SU(4) lattice model that is triangular at filling factor n=1 and honeycomb at filling factor n=2. The authors explore the phase diagram at n=1 a with the additional SU(4) breaking terms. At n=2, the authors concentrate on the SU(4)-symmetric model in the honeycomb lattice and leave an analysis of the symmetry-breaking for future work.

Because of the identification of the honeycomb model at n=2, I believe the manuscript is suitable for publication. However, this submission does not meet the criteria of SciPost Physics but does meet those of SciPost Physics Core, where it could be published.

Before acceptance, however, I would like to understand better the importance of considering the symmetry-breaking terms. The reason is two-fold: (1) the authors conjecture that the symmetry-allowed SOC is small; (2) in the caption of Fig. 9, the authors say that previous experiments did not access the regime where the symmetry-breaking terms become relevant.

Moreover, I have a series of questions that I believe would help to clarify some technical aspects of the work. I list them below:

1. How did the authors perform calculations that produced Fig. 3 and 10? Did they perform atomistic tight-binding calculations? Or do they discretize the continuum Hamiltonian in Eq. 1 on a lattice? Or maybe even the calculations performed analytically? If the plots result from numerical calculations, is the code used publicly available somewhere?
2. The authors conjecture a small layer-opposite Rashba spin-orbit coupling is small because they cannot imagine a microscopic origin for this term. Can they expand on the fundamental reason that prevents a large magnitude of LORSOC?
3. How did the authors calculate the phase diagram at n=1? Below the Eq. B5, they say that they minimized the classical Hamiltonian. What exactly does it mean? Did the authors find the minimal total energy varying the free parameters?
4. In the caption of Fig. 9, the authors point out that previous experimental works can only access the regions where $H_p/J_x$ is small. Can the authors envision a system where the upper part of the phase diagram should manifest? If not, does it mean that except for the small regions where the "umbrella" phase manifests, the SU(4)-symmetric model presented in Ref. 48 is sufficient to describe these AB twisted homobilayers?
5. To obtain the Wannier model at n=2, the authors compute maximally localized Wannier functions using a parametrized Gaussian function as the trial. Is it clear that the overlap between different orbitals is negligible?
6. Are the trial Wannier functions the same Gaussian as in Eq. C2 for the n=1 case?

Requested changes

1 - Provide more details on the calculations. Or make the code publicly available.
2 - Improve the resolution of the images.
3 - Address the comments in the report.

---

## Round 1 · Referee Report · Paul Eugenio (Referee 2) · 2023-5-24

Strengths

(1) Proposes SU(4) spin model on honeycomb lattice
(2) Relevant to current experimental investigations in TMD bilayers
(3) Very well resourced, with lots of relevant citations of theory and current experiment

Weaknesses

(1) Large focus on Rashba spin-orbit coupling, which is irrelevant for key results produced in the second half of the paper (i.e SU(4) honeycomb)
(2) Lacking details relevant to the Wannier functions used to construct tight-binding models

Report

The authors study WSe2 homobilayers marginally twisted away from an AB stacking. It is known that the AB stacking contributes opposite spins between layers within a valley, which suppresses the inter-layer tunneling due to symmetry. Therefore the device can be modeled by a disconnected pairs of layers, which at integer fillings and strong coupling simulates a certain class of spin model. The authors question if adding Rashba spin-orbit can mix the spins within a valley and thus generate an inter-layer tunneling, but find that its effects are marginal. They construct tight-binding models for single (per layer) and multi-orbital moire lattices, and find that the multi-orbital case exhibits an SU(4) honeycomb spin model (which to the best of my knowledge is novel).

To start, it is my expectation that Rashba spin-orbit is small in any moire system. This is because Rashba is proportional to the momentum, which for a moire band is cutoff by the moire BZ. Since the moire BZ size vanishes as the twist angle goes to zero, I similarly expect the same of Rashba.

Nevertheless, the question posed by the authors -- i.e can spin-mixing generate tunneling -- is a good one. However, I do not expect a vanishingly small Rashba to do this, especially given the large Ising spin-orbit, which itself prefers up/down spins.

In my opinion, this paper would gain by focusing more on the details of the tight-binding model which lead to the honeycomb spin system. At the very least this should involve plots of the Wannier functions.

Details pertaining to the small effects of the Rashba are not uninteresting, as it shows how an induced inter-layer tunneling can manifests as a spatially dependent field in the spin Hamiltonian; and I appreciate how Fig 8 tells us how such a tunneling can manifest as a breakdown in the layer-polarization plateau. But much of the discussion of Rashba could likely be relegated to the appendix, and most definitely shouldn't be the 3rd paragraph in the intro, as it creates a sense that it is the main result, or at least more so than the honeycomb. (Further, the intense focus on Rashba from the beginning gave me the initial sense that it was important for the honeycomb, only to find upon deeper reading that the authors have ignored Rashba for that calculation entirely. It thus detracts from the overall clarity.)

Despite its shortcomings, my overall opinion of this article is positive, as I have learned a lot by reading it. I do not request, but strongly suggest a restructuring of the story of the paper to focus on its more novel aspects.

I believe this article is fitting as a Scipost Physics Core article, and I recommend it as such.

Requested changes

(1) A very minor point: Correct the mispelling of "SU(4) Heisenberg mdoel" in the beginning of the fourth paragraph.
(2) Plots of the Wannier functions used for Eqn 2 & 6.

---

## Editorial Decision

awaiting_resubmission